# Direct Feedback Alignment Provides Learning in Deep Neural Networks

**Arild Nøkland**
Trondheim, Norway
`arild.nokland@gmail.com`

## Abstract

Artificial neural networks are most commonly trained with the back-propagation algorithm, where the gradient for learning is provided by back-propagating the error, layer by layer, from the output layer to the hidden layers. A recently discovered method called feedback-alignment shows that the weights used for propagating the error backward don't have to be symmetric with the weights used for propagation the activation forward. In fact, random feedback weights work evenly well, because the network learns how to make the feedback useful. In this work, the feedback alignment principle is used for training hidden layers more independently from the rest of the network, and from a zero initial condition. The error is propagated through fixed random feedback connections directly from the output layer to each hidden layer. This simple method is able to achieve zero training error even in convolutional networks and very deep networks, completely without error back-propagation. The method is a step towards biologically plausible machine learning because the error signal is almost local, and no symmetric or reciprocal weights are required. Experiments show that the test performance on MNIST and CIFAR is almost as good as those obtained with back-propagation for fully connected networks. If combined with dropout, the method achieves 1.45% error on the permutation invariant MNIST task.

## 1 Introduction

For supervised learning, the back-propagation algorithm (BP), see [2], has achieved great success in training deep neural networks. As today, this method has few real competitors due to its simplicity and proven performance, although some alternatives do exist.

Boltzmann machine learning in different variants are biologically inspired methods for training neural networks, see [6], [10] and [5]. The methods use only local available signals for adjusting the weights. These methods can be combined with BP fine-tuning to obtain good discriminative performance.

Contrastive Hebbian Learning (CHL), is similar to Boltzmann Machine learning, but can be used in deterministic feed-forward networks. In the case of weak symmetric feedback-connections it resembles BP [16].

Recently, target-propagation (TP) was introduced as an biologically plausible training method, where each layer is trained to reconstruct the layer below [7]. This method does not require symmetric weights and propagates target values instead of gradients backward.

A novel training principle called feedback-alignment (FA) was recently introduced [9]. The authors show that the feedback weights used to back-propagate the gradient do not have to be symmetric with the feed-forward weights. The network learns how to use fixed random feedback weights in order to reduce the error. Essentially, the network learns how to learn, and that is a really puzzling result.

Back-propagation with asymmetric weights was also explored in [8]. One of the conclusions from this work is that the weight symmetry constraint can be significantly relaxed while still retaining strong performance.

The back-propagation algorithm is not biologically plausible for several reasons. First, it requires symmetric weights. Second, it requires separate phases for inference and learning. Third, the learning signals are not local, but have to be propagated backward, layer-by-layer, from the output units. This requires that the error derivative has to be transported as a second signal through the network. To transport this signal, the derivative of the non-linearities have to be known.

All mentioned methods require the error to travel backward through reciprocal connections. This is biologically plausible in the sense that cortical areas are known to be reciprocally connected [3]. The question is how an error signal is relayed through an area to reach more distant areas. For BP and FA the error signal is represented as a second signal in the neurons participating in the forward pass. For TP the error is represented as a change in the activation in the same neurons. Consider the possibility that the error in the relay layer is represented by neurons not participating in the forward pass. For lower layers, this implies that the feedback path becomes disconnected from the forward path, and the layer is no longer reciprocally connected to the layer above.

The question arise whether a neuron can receive a teaching signal also through disconnected feedback paths. This work shows experimentally that directly connected feedback paths from the output layer to neurons earlier in the pathway is sufficient to enable error-driven learning in a deep network. The requirements are that the feedback is random and the whole network is adapted. The concept is quite different from back-propagation, but the result is very similar. Both methods seem to produce features that makes classification easier for the layers above.

Figure 1c) and d) show the novel feedback path configurations that is further explored in this work. The methods are based on the feedback alignment principle and is named "direct feedback-alignment" (DFA) and "indirect feedback-alignment" (IFA).

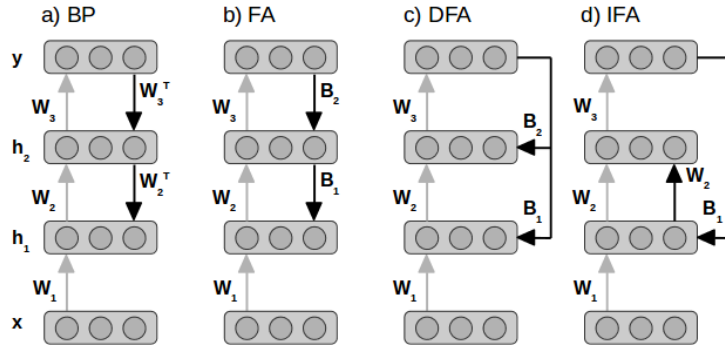

Figure 1: Overview of different error transportation configurations. Grey arrows indicate activation paths and black arrows indicate error paths. Weights that are adapted during learning are denoted as $W_i$, and weights that are fixed and random are denoted as $B_i$. a) Back-propagation. b) Feedback-alignment. c) Direct feedback-alignment. d) Indirect feedback-alignment.

## 2    Method

Let $(x, y)$ be mini-batches of input-output vectors that we want the network to learn. For simplicity, assume that the network has only two hidden layers as in Figure 1, and that the target output $y$ is scaled between 0 and 1. Let the rows in $W_i$ denote the weights connecting the layer below to a unit in hidden layer $i$, and let $b_i$ be a column vector with biases for the units in hidden layer $i$. The activations in the network are then calculated as

$$a_1 = W_1 x + b_1, \ h_1 = f(a_1) \tag{1}$$
$$a_2 = W_2 h_1 + b_2, \ h_2 = f(a_2) \tag{2}$$

$$a_y = W_3 h_2 + b_3, \ \hat{y} = f_y(a_y) \tag{3}$$

where $f()$ is the non-linearity used in hidden layers and $f_y()$ the non-linearity used in the output layer. If we choose a logistic activation function in the output layer and a binary cross-entropy loss function, the loss for a mini-batch with size $N$ and the gradient at the output layer $e$ are calculated as

$$J = -\frac{1}{N} \sum_{m,n} y_{mn} \log \hat{y}_{mn} + (1 - y_{mn}) \log(1 - \hat{y}_{mn}) \tag{4}$$

$$e = \delta a_y = \frac{\partial J}{\partial a_y} = \hat{y} - y \tag{5}$$

where $m$ and $n$ are output unit and mini-batch indexes. For the BP, the gradients for hidden layers are calculated as

$$\delta a_2 = \frac{\partial J}{\partial a_2} = (W_3^T e) \odot f'(a_2), \ \delta a_1 = \frac{\partial J}{\partial a_1} = (W_2^T \delta a_2) \odot f'(a_1) \tag{6}$$

where $\odot$ is an element-wise multiplication operator and $f'()$ is the derivative of the non-linearity. This gradient is also called steepest descent, because it directly minimizes the loss function given the linearized version of the network. For FA, the hidden layer update directions are calculated as

$$\delta a_2 = (B_2 e) \odot f'(a_2), \ \delta a_1 = (B_1 \delta a_2) \odot f'(a_1) \tag{7}$$

where $B_i$ is a fixed random weight matrix with appropriate dimension. For DFA, the hidden layer update directions are calculated as

$$\delta a_2 = (B_2 e) \odot f'(a_2), \ \delta a_1 = (B_1 e) \odot f'(a_1) \tag{8}$$

where $B_i$ is a fixed random weight matrix with appropriate dimension. If all hidden layers have the same number of neurons, $B_i$ can be chosen identical for all hidden layers. For IFA, the hidden layer update directions are calculated as

$$\delta a_2 = (W_2 \delta a_1) \odot f'(a_2), \ \delta a_1 = (B_1 e) \odot f'(a_1) \tag{9}$$

where $B_1$ is a fixed random weight matrix with appropriate dimension. Ignoring the learning rate, the weight updates for all methods are calculated as

$$\delta W_1 = -\delta a_1 x^T, \ \delta W_2 = -\delta a_2 h_1^T, \ \delta W_3 = -e h_2^T \tag{10}$$

## 3 Theoretical results

BP provides a gradient that points in the direction of steepest descent in the loss function landscape. FA provides a different update direction, but experimental results indicate that the method is able to reduce the error to zero in networks with non-linear hidden units. This is surprising because the principle is distinct different from steepest descent. For BP, the feedback weights are the transpose of the forward weights. For FA the feedback weights are fixed, but if the forward weights are adapted, they will approximately align with the pseudoinverse of the feedback weights in order to make the feedback useful [9].

The feedback-alignment paper [9] proves that fixed random feedback asymptotically reduces the error to zero. The conditions for this to happen are freely restated in the following. 1) The network is linear with one hidden layer. 2) The input data have zero mean and standard deviation one. 3) The feedback matrix $B$ satisfies $B^+ B = I$ where $B^+$ is the Moore-Penrose pseudo-inverse of $B$. 4) The forward weights are initialized to zero. 5) The output layer weights are adapted to minimize the error. Let's call this novel principle the feedback alignment principle.

It is not clear how the feedback alignment principle can be applied to a network with several non-linear hidden layers. The experiments in [9] show that more layers can be added if the error is back-propagated layer-by-layer from the output.

The following theorem points at a mechanism that can explain the feedback alignment principle. The mechanism explains how an asymmetric feedback path can provide learning by aligning the back-propagated and forward propagated gradients with it's own, under the assumption of constant update directions for each data point.

**Theorem 1.** *Given 2 hidden layers $k$ and $k+1$ in a feed-forward neural network where $k$ connects to $k+1$. Let $h_k$ and $h_{k+1}$ be the hidden layer activations. Let the functional dependency between the layers be $h_{k+1} = f(a_{k+1})$, where $a_{k+1} = Wh_k + b$. Here $W$ is a weight matrix, $b$ is a bias vector and $f()$ is a non-linearity. Let the layers be updated according to the non-zero update directions $\delta h_k$ and $\delta h_{k+1}$ where $\frac{\delta h_k}{\|\delta h_k\|}$ and $\frac{\delta h_{k+1}}{\|\delta h_{k+1}\|}$ are constant for each data point. The negative update directions will minimize the following layer-wise criterion*

$$K = K_k + K_{k+1} = \frac{\delta h_k^T h_k}{\|\delta h_k\|} + \frac{\delta h_{k+1}^T h_{k+1}}{\|\delta h_{k+1}\|} \tag{11}$$

*Minimizing $K$ will maximize the gradient maximizing the alignment criterion*

$$L = L_k + L_{k+1} = \frac{\delta h_k^T c_k}{\|\delta h_k\|} + \frac{\delta h_{k+1}^T c_{k+1}}{\|\delta h_{k+1}\|} \tag{12}$$

*where*

$$c_k = \frac{\partial h_{k+1}}{\partial h_k} \delta h_{k+1} = W^T(\delta h_{k+1} \odot f'(a_{k+1})) \tag{13}$$

$$c_{k+1} = \frac{\partial h_{k+1}}{\partial h_k^T} \delta h_k = (W\delta h_k) \odot f'(a_{k+1}) \tag{14}$$

*If $L_k > 0$, then is $-\delta h_k$ a descending direction in order to minimize $K_{k+1}$.*

*Proof.* Let $i$ be the any of the layers $k$ or $k+1$. The prescribed update $-\delta h_i$ is the steepest descent direction in order to minimize $K_i$ because by using the product rule and the fact that any partial derivative of $\frac{\delta h_i}{\|\delta h_i\|}$ is zero we get

$$-\frac{\partial K_i}{\partial h_i} = -\frac{\partial}{\partial h_i}\left[\frac{\delta h_i^T h_i}{\|\delta h_i\|}\right] = -\frac{\partial}{\partial h_i}\left[\frac{\delta h_i}{\|\delta h_i\|}\right]h_i - \frac{\partial h_i}{\partial h_i}\frac{\delta h_i}{\|\delta h_i\|} = -0h_i - \frac{\delta h_i}{\|\delta h_i\|} = -\alpha_i \delta h_i \tag{15}$$

Here $\alpha_i = \frac{1}{\|\delta h_i\|}$ is a positive scalar because $\delta h_i$ is non-zero. Let $\delta a_i$ be defined as $\delta a_i = \frac{\partial h_i}{\partial a_i}\delta h_i = \delta h_i \odot f'(a_i)$ where $a_i$ is the input to layer $i$. Using the product rule again, the gradients maximizing $L_k$ and $L_{k+1}$ are

$$\frac{\partial L_i}{\partial c_i} = \frac{\partial}{\partial c_i}\left[\frac{\delta h_i^T c_i}{\|\delta h_i\|}\right] = \frac{\partial}{\partial c_i}\left[\frac{\delta h_i}{\|\delta h_i\|}\right]c_i + \frac{\partial c_i}{\partial c_i}\frac{\delta h_i}{\|\delta h_i\|} = 0c_i + \frac{\delta h_i}{\|\delta h_i\|} = \alpha_i \delta h_i \tag{16}$$

$$\frac{\partial L_{k+1}}{\partial W} = \frac{\partial L_{k+1}}{\partial c_{k+1}}\frac{\partial c_{k+1}}{\partial W} = \alpha_{k+1}(\delta h_{k+1} \odot f'(a_{k+1}))\delta h_k^T = \alpha_{k+1}\delta a_{k+1}\delta h_k^T \tag{17}$$

$$\frac{\partial L_k}{\partial W} = \frac{\partial c_k}{\partial W^T}\frac{\partial L_k}{\partial c_k^T} = (\delta h_{k+1} \odot f'(a_{k+1}))\alpha_k \delta h_k^T = \alpha_k \delta a_{k+1}\delta h_k^T \tag{18}$$

Ignoring the magnitude of the gradients we have $\frac{\partial L}{\partial W} = \frac{\partial L_k}{\partial W} = \frac{\partial L_{k+1}}{\partial W}$. If we project $h_i$ onto $\delta h_i$ we can write $h_i = \frac{h_i^T \delta h_i}{\|\delta h_i\|^2}\delta h_i + h_{i,res} = \alpha_i K_i \delta h_i + h_{i,res}$. For $W$, the prescribed update is

$$\delta W = -\delta h_{k+1}\frac{\partial h_{k+1}}{\partial W} = -(\delta h_{k+1} \odot f'(a_{k+1}))h_k^T = -\delta a_{k+1}h_k^T = -\delta a_{k+1}(\alpha_k K_k \delta h_k + h_{k,res})^T =$$

$$-\alpha_k K_k \delta a_{k+1}\delta h_k^T - \delta a_{k+1}h_{k,res}^T = -K_k\frac{\partial L_k}{\partial W} - \delta a_{k+1}h_{k,res}^T \tag{19}$$

We can indirectly maximize $L_k$ and $L_{k+1}$ by maximizing the component of $\frac{\partial L_k}{\partial W}$ in $\delta W$ by minimizing $K_k$. The gradient to minimize $K_k$ is the prescribed update $-\delta h_k$.

$L_k > 0$ implies that the angle $\beta$ between $\delta h_k$ and the back-propagated gradient $c_k$ is within $90°$ of each other because $cos(\beta) = \frac{c_k^T \delta h_k}{\|c_k\|\|\delta h_k\|} = \frac{L_k}{\|c_k\|} > 0 \Rightarrow |\beta| < 90°$. $L_k > 0$ also implies that $c_k$ is non-zero and thus descending. Then $\delta h_k$ will point in a descending direction because a vector within $90°$ of the steepest descending direction will also point in a descending direction.

$\square$

It is important to note that the theorem doesn't tell that the training will converge or reduce any error to zero, but if the fake gradient is successful in reducing $K$, then will this gradient also include a growing component that tries to increase the alignment criterion $L$.

The theorem can be applied to the output layer and the last hidden layer in a neural network. To achieve error-driven learning, we have to close the feedback loop. Then we get the update directions $\delta h_{k+1} = \frac{\partial J}{\partial a_y} = e$ and $\delta h_k = G_k(e)$ where $G_k(e)$ is a feedback path connecting the output to the hidden layer. The prescribed update will directly minimize the loss $J$ given $h_k$. If $L_k$ turns positive, the feedback will provide a update direction $\delta h_k = G_k(e)$ that reduces the same loss. The theorem can be applied successively to deeper layers. For each layer $i$, the weight matrix $W_i$ is updated to minimize $K_{i+1}$ in the layer above, and at the same time indirectly make it's own update direction $\delta h_i = G_i(e)$ useful.

Theorem 1 suggests that a large class of asymmetric feedback paths can provide a descending gradient direction for a hidden layer, as long as on average $L_i > 0$. Choosing feedback paths $G_i(e)$, visiting every layer on it's way backward, with weights fixed and random, gives us the FA method. Choosing direct feedback paths $G_i(e) = B_i e$, with $B_i$ fixed and random, gives us the DFA method. Choosing a direct feedback path $G_1(e) = B_1 e$ connecting to the first hidden layer, and then visiting every layer on it's way forward, gives us the IFA method. The experimental section shows that learning is possible even with indirect feedback like this.

Direct random feedback $\delta h_i = G_i(e) = B_i e$ has the advantage that $\delta h_i$ is non-zero for all non-zero $e$. This is because a random matrix $B_i$ will have full rank with a probability very close to 1. A non-zero $\delta h_i$ is a requirement in order to achieve $L_i > 0$. Keeping the feedback static will ensure that this property is preserved during training. In addition, a static feedback can make it easier to maximize $L_i$ because the direction of $\delta h_i$ is more constant. If the cross-entropy loss is used, and the output target values are 0 or 1, then the sign of the error $e_j$ for a given sample $j$ will not change. This means that the quantity $B_i\, sign(e_j)$ will be constant during training because both $B_i$ and $sign(e_j)$ are constant. If the task is to classify, the quantity will in addition be constant for all samples within a class. Direct random feedback will also provide a update direction $\delta h_i$ with a magnitude that only varies with the magnitude of the error $e$.

If the forward weights are initialized to zero, then will $L_i = 0$ because the back-propagated error is zero. This seems like a good starting point when using asymmetric feedback because the first update steps have the possibility to quickly turn this quantity positive. A zero initial condition is however not a requirement for asymmetric feedback to work. One of the experiments will show that even when starting from a bad initial condition, direct random and static feedback is able to turn this quantity positive and reduce the training error to zero.

For FA and BP, the hidden layer growth is bounded by the layers above. If the layers above saturate, the hidden layer update $\delta h_i$ becomes zero. For DFA, the hidden layer update $\delta h_i$ will be non-zero as long as the error $e$ is non-zero. To limit the growth, a squashing non-linearity like hyperbolic tangent or logistic sigmoid seems appropriate. If we add a tanh non-linearity to the hidden layer, the hidden activation is bounded within $[-1, 1]$. With zero initial weights, $h_i$ will be zero for all data points. The tanh non-linearity will not limit the initial growth in any direction. The experimental results indicate that this non-linearity is well suited together with DFA.

If the hyperbolic tangent non-linearity is used in the hidden layer, the forward weights can be initialized to zero. The rectified linear activation function (ReLU) will not work with zero initial weights because the error derivative for such a unit is zero when the bias and incoming weights are all zero.

## 4 Experimental results

To investigate if DFA learns useful features in the hidden layers, a 3x400 tanh network was trained on MNIST with both BP and DFA. The input test images and resulting features were visualized using t-SNE [15], see Figure 3. Both methods learns features that makes it easier to discriminate between the classes. At the third hidden layer, the clusters are well separated, except for some stray points. The visible improvement in separation from input to first hidden layer indicates that error DFA is able to learn useful features also in deeper hidden layers.

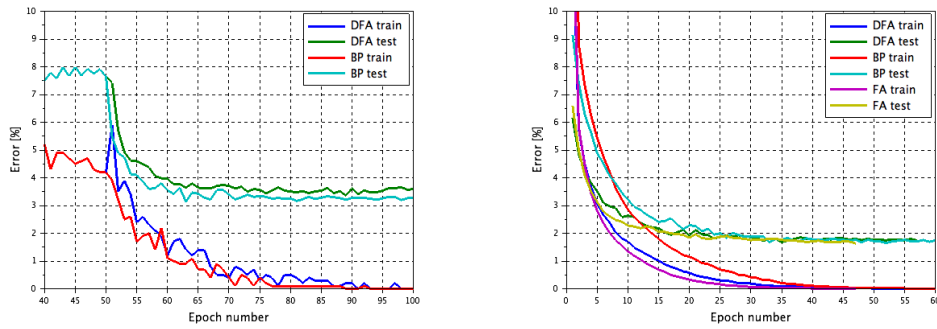

Figure 2: Left: Error curves for a network pre-trained with a frozen first hidden layer. Right: Error curves for normal training of a 2x800 tanh network on MNIST.

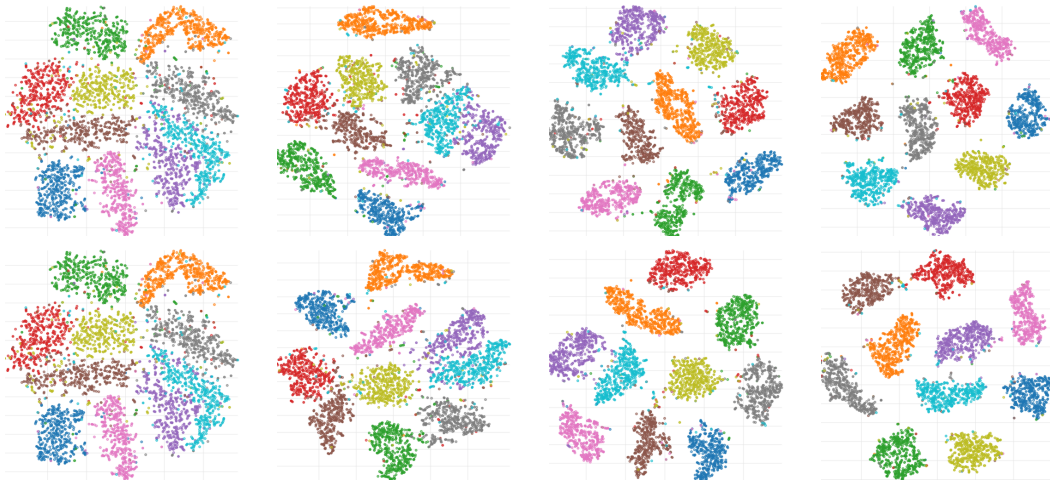

Figure 3: t-SNE visualization of MNIST input and features. Different colors correspond to different classes. The top row shows features obtained with BP, the bottom row shows features obtained with DFA. From left to right: input images, first hidden layer features, second hidden layer features and third hidden layer features.

Furthermore, another experiment was performed to see if error DFA is able to learn useful hidden representations in deeper layers. A 3x50 tanh network was trained on MNIST. The first hidden layer was fixed to random weights, but the 2 hidden layers above were trained with BP for 50 epochs. At this point, the training error was about 5%. Then, the first hidden layer was unfrozen and training continued with BP. The training error decreased to 0% in about 50 epochs. The last step was repeated, but this time the unfreezed layer was trained with DFA. As expected because of different update directions, the error first increased, then decreased to 0% after about 50 epochs. The error curves are presented in Figure2(Left). Even though the update direction provided by DFA is different from the back-propagated gradient, the resulting hidden representation reduces the error in a similar way.

Several feed-forward networks were trained on MNIST and CIFAR to compare the performance of DFA with FA and BP. The experiments were performed with the binary cross-entropy loss and optimized with RMSprop [14]. For the MNIST dropout experiments, learning rate with decay and training time was chosen based on a validation set. For all other experiments, the learning rate was roughly optimized for BP and then used for all methods. The learning rate was constant for each dataset. Training was stopped when training error reached 0.01% or the number of epochs reached 300. A mini-batch size of 64 was used. No momentum or weight decay was used. The input data was scaled to be between 0 and 1, but for the convolutional networks, the data was whitened. For FA and DFA, the weights and biases were initialized to zero, except for the ReLU networks. For BP and/or ReLU, the initial weights and biases were sampled from a uniform distribution in the range

$[-1/\sqrt{fanin}, 1/\sqrt{fanin}]$. The random feedback weights were sampled from a uniform distribution in the range $[-1/\sqrt{fanout}, 1/\sqrt{fanout}]$.

| MODEL | BP | FA | DFA |
|---|---|---|---|
| 7x240 Tanh | $2.16 \pm 0.13\%$ | $2.20 \pm 0.13\% \,(0.02\%)$ | $2.32 \pm 0.15\% \,(0.03\%)$ |
| 100x240 Tanh | | | $3.92 \pm 0.09\% \,(0.12\%)$ |
| 1x800 Tanh | $1.59 \pm 0.04\%$ | $1.68 \pm 0.05\%$ | $1.68 \pm 0.05\%$ |
| 2x800 Tanh | $1.60 \pm 0.06\%$ | $1.64 \pm 0.03\%$ | $1.74 \pm 0.08\%$ |
| 3x800 Tanh | $1.75 \pm 0.05\%$ | $1.66 \pm 0.09\%$ | $1.70 \pm 0.04\%$ |
| 4x800 Tanh | $1.92 \pm 0.11\%$ | $1.70 \pm 0.04\%$ | $1.83 \pm 0.07\% \,(0.02\%)$ |
| 2x800 Logistic | $1.67 \pm 0.03\%$ | $1.82 \pm 0.10\%$ | $1.75 \pm 0.04\%$ |
| 2x800 ReLU | $1.48 \pm 0.06\%$ | $1.74 \pm 0.10\%$ | $1.70 \pm 0.06\%$ |
| 2x800 Tanh + DO | $1.26 \pm 0.03\% \,(0.18\%)$ | $1.53 \pm 0.03\% \,(0.18\%)$ | $1.45 \pm 0.07\% \,(0.24\%)$ |
| 2x800 Tanh + ADV | $1.01 \pm 0.08\%$ | $1.14 \pm 0.03\%$ | $1.02 \pm 0.05\% \,(0.12\%)$ |

Table 1: MNIST test error for back-propagation (BP), feedback-alignment (FA) and direct feedback-alignment (DFA). Training error in brackets when higher than 0.01%. Empty fields indicate no convergence.

The results on MNIST are summarized in Table 1. For adversarial regularization (ADV), the networks were trained on adversarial examples generated by the "fast-sign-method" [4]. For dropout regularization (DO) [12], a dropout probability of $0.1$ was used in the input layer and $0.5$ elsewhere. For the 7x240 network, target propagation achieved an error of 1.94% [7]. The results for all three methods are very similar. Only DFA was able to train the deepest network with the simple initialization used. The best result for DFA matches the best result for BP.

| MODEL | BP | FA | DFA |
|---|---|---|---|
| 1x1000 Tanh | $45.1 \pm 0.7\% \,(2.5\%)$ | $46.4 \pm 0.4\% \,(3.2\%)$ | $46.4 \pm 0.4\% \,(3.2\%)$ |
| 3x1000 Tanh | $45.1 \pm 0.3\% \,(0.2\%)$ | $47.0 \pm 2.2\% \,(0.3\%)$ | $47.4 \pm 0.8\% \,(2.3\%)$ |
| 3x1000 Tanh + DO | $42.2 \pm 0.2\% \,(36.7\%)$ | $46.9 \pm 0.3\% \,(48.9\%)$ | $42.9 \pm 0.2\% \,(37.6\%)$ |
| CONV Tanh | $22.5 \pm 0.4\%$ | $27.1 \pm 0.8\% \,(0.9\%)$ | $26.9 \pm 0.5\% \,(0.2\%)$ |

Table 2: CIFAR-10 test error for back-propagation (BP), feedback-alignment (FA) and direct feedback-alignment (DFA). Training error in brackets when higher than 0.1%.

The results on CIFAR-10 are summarized in Table 2. For the convolutional network the error was injected after the max-pooling layers. The model was identical to the one used in the dropout paper [12], except for the non-linearity. For the 3x1000 network, target propagation achieved an error of 49.29% [7]. For the dropout experiment, the gap between BP and DFA is only 0.7%. FA does not seem to improve with dropout. For the convolutional network, DFA and FA are worse than BP.

| MODEL | BP | FA | DFA |
|---|---|---|---|
| 1x1000 Tanh | $71.7 \pm 0.2\% \,(38.7\%)$ | $73.8 \pm 0.3\% \,(37.5\%)$ | $73.8 \pm 0.3\% \,(37.5\%)$ |
| 3x1000 Tanh | $72.0 \pm 0.3\% \,(0.2\%)$ | $75.3 \pm 0.1\% \,(0.5\%)$ | $75.9 \pm 0.2\% \,(3.1\%)$ |
| 3x1000 Tanh + DO | $69.8 \pm 0.1\% \,(66.8\%)$ | $75.3 \pm 0.2\% \,(77.2\%)$ | $73.1 \pm 0.1\% \,(69.8\%)$ |
| CONV Tanh | $51.7 \pm 0.2\%$ | $60.5 \pm 0.3\%$ | $59.0 \pm 0.3\%$ |

Table 3: CIFAR-100 test error for back-propagation (BP), feedback-alignment (FA) and direct feedback-alignment (DFA). Training error in brackets when higher than 0.1%.

The results on CIFAR-100 are summarized in Table 3. DFA improves with dropout, while FA does not. For the convolutional network, DFA and FA are worse than BP.

The above experiments were performed to verify the DFA method. The feedback loops are the shortest possible, but other loops can also provide learning. An experiment was performed on MNIST

to see if a single feedback loop like in Figure 1d), was able to train a deep network with 4 hidden layers of 100 neurons each. The feedback was connected to the first hidden layer, and all hidden layers above were trained with the update direction forward-propagated through this loop. Starting from a random initialization, the training error reduced to 0%, and the test error reduced to 3.9%.

## 5  Discussion

The experimental results indicate that DFA is able to fit the training data equally good as BP and FA. The performance on the test set is similar to FA but lagging a little behind BP. For the convolutional network, BP is clearly the best performer. Adding regularization seems to help more for DFA than for FA.

Only DFA was successful in training a network with 100 hidden layers. If proper weight initialization is used, BP is able to train very deep networks as well [13][11]. The reason why BP fails to converge is probably the very simple initialization scheme used here. Proper initialization might help FA in a similar way, but this was not investigated any further.

The DFA training procedure has a lot in common with supervised layer-wise pre-training of a deep network, but with an important difference. If all layers are trained simultaneously, it is the error at the top of a deep network that drives the learning, not the error in a shallow pre-training network.

If the network above a target hidden layer is not adapted, FA and DFA will not give an improvement in the loss. This is in contrast to BP that is able to decrease the error even in this case because the feedback depends on the weights and layers above.

DFA demonstrates a novel application of the feedback alignment principle. The brain may or may not implement this kind of feedback, but it is a step towards better better understanding mechanisms that can provide error-driven learning in the brain. DFA shows that learning is possible in feedback loops where the forward and feedback paths are disconnected. This introduces a large flexibility in how the error signal might be transmitted. A neuron might receive it's error signals via a post-synaptic neuron (BP,CHL), via a reciprocally connected neuron (FA,TP), directly from a pre-synaptic neuron (DFA), or indirectly from an error source located several synapses away earlier in the informational pathway (IFA).

Disconnected feedback paths can lead to more biologically plausible machine learning. If the feedback signal is added to the hidden layers before the non-linearity, the derivative of the non-linearity does not have to be known. The learning rule becomes local because the weight update only depends on the pre-synaptic activity and the temporal derivative of the post-synaptic activity. Learning is not a separate phase, but performed at the end of an extended forward pass. The error signal is not a second signal in the neurons participating in the forward pass, but a separate signal relayed by other neurons. The local update rule can be linked to Spike-Timing-Dependent Plasticity (STDP) believed to govern synaptic weight updates in the brain, see [1].

Disconnected feedback paths have great similarities with controllers used in dynamical control loops. The purpose of the feedback is to provide a change in the state that reduces the output error. For a dynamical control loop, the change is added to the state and propagated forward to the output. For a neural network, the change is used to update the weights.

## 6  Conclusion

A biologically plausible training method based on the feedback alignment principle is presented for training neural networks with error feedback rather than error back-propagation. In this method, neither symmetric weights nor reciprocal connections are required. The error paths are short and enables training of very deep networks. The training signals are local or available at most one synapse away. No weight initialization is required.

The method was able to fit the training set on all experiments performed on MNIST, Cifar-10 and Cifar-100. The performance on the test sets lags a little behind back-propagation.

Most importantly, this work suggests that the restriction enforced by back-propagation and feedback-alignment, that the backward pass have to visit every neuron from the forward pass, can be discarded. Learning is possible even when the feedback path is disconnected from the forward path.

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
