[Reviews · NeurIPS 2016]

Reviewer 1

Summary

This paper demonstrates empirically that a remarkably simple feedback scheme can successfully train very deep networks. In particular, the paper shows that the error signal can be directly propagated to each hidden layer through a fixed, random weight matrix. This differs from backpropagation and the recently-proposed feedback alignment scheme in that there is no backward flow of error information through layers. Instead, error information is sent in one step to each layer. Remarkably, simulations show that this scheme succeeds in training deep networks on the MNIST and CIFAR-10 datasets, with comparable speed and accuracy as backpropagation and feedback alignment. This scheme is clearly biologically plausible: error information is broadcast in one step to all layers through random connections.

Qualitative Assessment

This paper demonstrates the surprising result that error information need not flow backward through several layers to enable good learning in a deep network. The scheme proposed in the paper is definitely biologically plausible (which is not to say that the brain necessarily uses this scheme). The paper could be strengthened through a more complete analysis of the learned filters in lower layers. Do they still visually resemble the filters produced through backpropagation? In particular, it seems important to understand why DFA attains the same training error as BP, but worse test error (particularly on CIFAR). Do DFA solutions have higher norm weights as compared to BP? Do DFA solutions change weights less or more in lower vs higher layers? Analyses of this sort may uncover the type of implicit regularization at work in DFA. It seems highly significant that the method obtains good training performance but slightly worse testing performance. The theoretical results are more intuitions. None of the statements cover the actual DFA scheme proposed in the paper. The discussion contains several statements that might be true but which are not backed up by evidence. For instance, that random feedback can help escape local minima in which standard gradient descent becomes trapped. There is little evidence that gradient descent becomes trapped in a local minimum (Saxe et al., 2014; Dauphin et al, 2014; Choromanska et al., 2015; Goodfellow et al., 2015) and no clear reason to suspect that random feedback or DFA will not. I am not convinced that DFA works better than backpropagation for very deep networks—it likely depends on the weight initialization (eg Saxe et al., 2014). Several authors have reported very fast backpropagation-based training of even 1000 layer tanh networks with appropriate initializations (Sussillo & Abbott, 2014). The fact that the 100 layer tanh networks failed to converge to zero error in this work thus suggests that they were improperly initialized or optimized.

Confidence in this Review

3-Expert (read the paper in detail, know the area, quite certain of my opinion)


Reviewer 2

Summary

Based on previous work (Feedback Alignment; Lillicrap et. al 2014) the authors here propose a new training method to replace backpropagation for training deep (supervised) neural networks. In contrast to previous work, the authors here show that individual feedback loops between the layers and the final loss can be used to optimize all layers of a feed-forward network.

Qualitative Assessment

The motivation and introduction of the basic idea and the actual proofs are well presented. At first glance the proposed method looks like a simple and straightforward extension t(actually: simplification) of previous work on feedback alignment; but the proposed changes and their impact could be significant if the optimization method would lead similar good solutions compared to backprop. Especially because the proposed method could potentially lead to more biologically plausible neural network architectures / methods. The experimental section is extensive and the results indicate that the proposed method indeed archives competitive results. I think the presented results are very interesting and sufficient to warrant a publication. But they are not conclusive to belief DFA indeed archives backprop-like performance over a large variety of tasks. The experimental section also contains many details and contains probably all the necessary information to reproduce the experiments.

Confidence in this Review

2-Confident (read it all; understood it all reasonably well)


Reviewer 3

Summary

Feedback alignment (FA) is a recently developed method for training neural networks which is more biologically plausible than backpropagation (BP. Unlike BP, where the error signal is propagated on the backward pass through the transpose of the forward weights (which is difficult to square with our understanding of neural constraints), in feedback alignment a random weight matrix is used to propagate the error. Surprisingly, propagating the error though random weights works well, learns quickly (although sometimes slightly slower than BP) and may even act as a regularizer and outperform backpropagation in some situations. This paper proposes, analyses and experimentally tests a variant of this approach they term "direct feedback alignment." In this scheme, rather than propagating the error through random weights at each layer, the error signal for each layer is derived through random ways directly from the output layer. The find that this scheme is able to perform similar to FA, and unlike both BP and FA is capable of training very deep networks.

Qualitative Assessment

This paper is relatively well-written and supports its claims. It develops a reasonable analysis of the direct feedback alignment, although it follows closely from the original feedback alignment work. For this reason, it would probably be possible to abbreviate parts of the analysis without much loss to the reader. The primary weakness of this work is impact. The idea of "direct feedback alignment" follows fairly straightforwardly from the original FA alignment work. Its notable that it is useful in training very deep networks (e.g. 100 layers) but its not clear that this results in an advantage for function approximation (the error rate is higher for these deep networks). If the authors could demonstrate that DFA allows one to train and make use of such deep networks where BP and FA struggle on a larger dataset this would significantly enhance the impact of the paper. In terms of biological understanding, FA seems more supported by biological observations (which typically show reciprocal forward and backward connections between hierarchical brain areas, not direct connections back from one region to all others as might be expected in DFA). The paper doesn't provide support for their claim, in the final paragraph, that DFA is more biologically plausible than FA. Minor issues: - A few typos, there is no line numbers in the draft so I haven't itemized them. - Table 1, 2, 3 the legends should be longer and clarify whether the numbers are % errors, or % correct (MNIST and CIFAR respectively presumably). - Figure 2 right. I found it difficult to distinguish between the different curves. Maybe make use of styles (e.g. dashed lines) or add color. - Figure 3 is very hard to read anything on the figure. - I think this manuscript is not following the NIPS style. The citations are not by number and there are no line numbers or an "Anonymous Author" placeholder. - I might be helpful to quantify and clarify the claim "ReLU does not work very well in very deep or in convolutional networks." ReLUs were used in the AlexNet paper which, at the time, was considered deep and makes use of convolution (with pooling rather than ReLUs for the convolutional layers).

Confidence in this Review

2-Confident (read it all; understood it all reasonably well)


Reviewer 4

Summary

The authors build upon the feedback alignment (FA) technique by Lillicrap et al. and investigate a version of FA, called direct feedback alignment (DFA), in which each (asymmetric) alignment matrix directly connects the error signal (wrt the output layer) to a hidden layer. Some theoretical justifications are given and some experiments were performed. In the latter, the authors found that DFA performance is almost on par, but slightly lags behind that of BP on most feedforward architectures when tested on MNIST, CFAR-10, and CFAR-100 datasets. However, very deep networks seem to benefit more from DFA than BP.

Qualitative Assessment

Every section of the paper was well explained except for the theoretical considerations, which reads like the writing of someone entirely distinct from the authors of any other section. There are small mistakes, which will be elaborated below, that come together to make the reader of that section very confused, but there are also larger issues, for example stating theorems verbally when doing so mathematically would be much more clear, and also more in line with the tone set up in the Method section. In addition, some graphics could have cleared up some of the definitions and theorem statements. These can be put in an appendix to fit the paper length limit. The following are some specific comments in line order. 1. In section 3, the clause "For a linear neural network with one hidden layer" should be listed as condition 1. I was confused below where the text refers to the "linearity assumption". It's only after I looked at the Lillicrap et al. paper that I understood the intent here. 2. By "The input units" do you mean data? (I believe this is what is below referred to as the "condition on the data") 3. "The zero weight condition is easy to satisfy if the hyperbolic tangent non-linearity is used in the hidden layers." I think I understand what you want to say, that you want to approximate the conditions for linear neural net stated in Lillicrap et al. in the setting of neural net with nonlinear transfer functions. But without having read Lillicrap, this sentence is very confusing. 4. "Then the gradient will be non-zero even if the weights start at zero." Why does gradient matter if we are talking about FA not BP? Again, I think you want to say that the tanh transfer function can still approximate linear transfer when weights start at zero. If so, your explanation is somewhat cryptic, that was undecipherable to me until I read Lillicrap et al. 5. "If the layers above saturates, the gradient be zero." s/saturates/saturate/, s/be/becomes/ 6. s/origo/the origin/ 7. "This is in contradiction to BP." s/contradiction/contrast/ 8. Definition 3.2. e and e_p are confusing. Is e the error at the output, and e_p the error at hidden state? Perhaps an example or some graphics would be illuminating. 9. s/infitesimal/infinitesimal/ 10. Lemma 1. Please use precise mathematical formulation. It's not very clear what you are proving here. 11. "Note that if the length of ep is bounded." s/length/norm/ 12. Lemma 2. Isn't this just backpropagation? What are you trying to say here?

Confidence in this Review

2-Confident (read it all; understood it all reasonably well)


Reviewer 5

Summary

This paper proposes to propagate the error through fixed random feedback connections directly from the output layer to each hidden layer. The theory shows that the feedback loop will provide a gradient in a descending direction unless the error is zero. The experimental results show that the proposed method can achieve better training performance but bad test performance compared with traditional backpropagation.

Qualitative Assessment

1. The experimental results in the paper cannot demonstrate the advantage or the vaule of the proposed DFA algorithm. The experimental results show that the DFA achieves very good training performance (e.g., almost zero training error) but bad test performance compared with BP and FA. These results only show the disadvantage of the proposed DFA, i.e., the proposed DFA is easier to overfit to the training data ant fall in bad local optima. 2. Important baseline is missing for the experiments for CIFAR-10 and CIFAR-100. These experiments should also be compared with FA. 3. The theoretical part is difficult to understand. It would be better to give some formal notations and use mathematical description.

Confidence in this Review

2-Confident (read it all; understood it all reasonably well)


Reviewer 6

Summary

Authors proposed to use DFA (direct feedback-alignment), a 'biologically plausible method', and compared its performance with BP (Back-propagation) and FA (feedback-alignment) on MNIST and CIFAR data. They argued that using directly connected feedback paths from the output layers to hidden layer is sufficient in error-driven learning, and showed the similarities between error feedback paths and dynamical feedback control loops. Their results suggested that DFA had slightly worse performance in MNIST and CIFAR-10, and only better than BP in CIFAR-100 3x1000 network without dropout.

Qualitative Assessment

Authors proposed a more 'biologically plausible training method' that does not have the restriction of symmetric weights or reciprocal connections as in BP and showed that it could fit the training set on MNIST, Cifar-10 and Cifar-100 but had lower performance on test set than BP. Overall it is an interesting read. Though it would be great if authors could also show the results with FA in CIFAR data, and be more clearer of the difference/improvement between FA and DFA. Minor points: - please make color legend consistent between Figure 2 Left and Right.

Confidence in this Review

2-Confident (read it all; understood it all reasonably well)